# Information Dynamics of the Brain, Cardiovascular and Respiratory Network during Different Levels of Mental Stress

**DOI:** 10.3390/e21030275

**Published:** 2019-03-13

**Authors:** Matteo Zanetti, Luca Faes, Giandomenico Nollo, Mariolino De Cecco, Riccardo Pernice, Luca Maule, Marco Pertile, Alberto Fornaser

**Affiliations:** 1Department of Industrial Engineering, University of Trento, 38123 Trento, Italy; 2Department of Engineering, University of Palermo, 90133 Palermo, Italy; 3Department of Industrial Engineering, University of Padova, 35131 Padova, Italy

**Keywords:** network physiology, information dynamics, wearable devices, stress assessment

## Abstract

In this study, an analysis of brain, cardiovascular and respiratory dynamics was conducted combining information-theoretic measures with the Network Physiology paradigm during different levels of mental stress. Starting from low invasive recordings of electroencephalographic, electrocardiographic, respiratory, and blood volume pulse signals, the dynamical activity of seven physiological systems was probed with one-second time resolution measuring the time series of the δ, θ, α and β brain wave amplitudes, the cardiac period (RR interval), the respiratory amplitude, and the duration of blood pressure wave propagation (pulse arrival time, PAT). Synchronous 5-min windows of these time series, obtained from 18 subjects during resting wakefulness (REST), mental stress induced by mental arithmetic (MA) and sustained attention induced by serious game (SG), were taken to describe the dynamics of the nodes composing the observed physiological network. Network activity and connectivity were then assessed in the framework of information dynamics computing the new information generated by each node, the information dynamically stored in it, and the information transferred to it from the other network nodes. Moreover, the network topology was investigated using directed measures of conditional information transfer and assessing their statistical significance. We found that all network nodes dynamically produce and store significant amounts of information, with the new information being prevalent in the brain systems and the information storage being prevalent in the peripheral systems. The transition from REST to MA was associated with an increase of the new information produced by the respiratory signal time series (RESP), and that from MA to SG with a decrease of the new information produced by PAT. Each network node received a significant amount of information from the other nodes, with the highest amount transferred to RR and the lowest transferred to δ, θ, α and β. The topology of the physiological network underlying such information transfer was node- and state-dependent, with the peripheral subnetwork showing interactions from RR to PAT and between RESP and RR, PAT consistently across states, the brain subnetwork resulting more connected during MA, and the subnetwork of brain–peripheral interactions involving different brain rhythms in the three states and resulting primarily activated during MA. These results have both physiological relevance as regards the interpretation of central and autonomic effects on cardiovascular and respiratory variability, and practical relevance as regards the identification of features useful for the automatic distinction of different mental states.

## 1. Introduction

Physiological systems react individually and interact among them in different manners during different physiological, cognitive and pathological states [1]. As a result, the physiological quantities constituting the output variables of the different physiological systems display a rich oscillatory activity, which is typically investigated through the acquisition of physiological signals obtained via low invasive instrumentation. These signals are elaborated to extract time series of interest, which are then analyzed using proper signal processing methods to reveal the underlying physiological mechanisms. The traditional approach consists of studying the function of the single system in isolation to prove the evidence of a relation between a particular property of a time series and a given physiological state. In the literature, some examples of this approach are the study of the heart rate variability (HRV) [2], respiratory variability [3], EEG power spectrum analysis [4,5] and electromyography (EMG) [6]. Another approach is the so-called multivariate approach, whereby multiple physiological signals are analyzed at the same time in order to extract information of interest from the dynamics of each individual signal or from the interaction between different signals. This approach is the basis of research fields investigating the distributed dynamical activity of individual organ systems, such as brain connectivity [7] and cardiac mapping [8], or the joint dynamical activity of anatomically connected systems, such as cardiovascular variability [9]. From an engineering point of view, this kind of approach is often used to extract features to train classifiers for the detection of particular physiological/mental states. Examples are studies concerning emotional recognition [10,11] or stress detection [12,13,14].

The multivariate approach to the study of physiological dynamics has moved a step forward with the recent introduction of the concept of Network Physiology [15,16]. With this perspective, the various physiological systems that compose the human organism are considered as nodes of a complex network. Accordingly, each system has its own internal regulatory mechanisms but also continuously interact with the other systems to assure correct responses to the various stimuli to achieve the proper functioning of the whole organism. This new paradigm requires that not only are quantities relevant to the dynamics of a single system studied, but also the interaction between different network nodes must be assessed in order to provide a thorough characterization of the human organism as a whole. To accomplish such an exhaustive description of the dynamical activity of the human physiological network, methods are needed which are able to deal with the diversity of the network nodes and the complexity of the resulting dynamics. In fact, different nodes of the human physiological network produce information at different rates, and the information produced can be preserved to a different extent for different nodes, or exchanged between nodes following multiple interaction pathways. In this case, since traditional analysis in time or frequency domains may not suffice, alternative approaches coming from the information domain can be proposed to investigate multiple aspects of the dynamics of physiological networks. In particular, the framework of Information Dynamics has been recently introduced to quantify, from multivariate time series representing the dynamical activity of multiple systems, the amounts of information produced by each system, stored in the system, transferred to it from the other connected systems, and modified as a consequence of the interaction between source systems sending information to a target system [17,18]. Measures taken from this framework have been used recently to study physiological brain–heart interactions during sleep [19,20] and during visual emotional elicitation [21], and to assess cardiovascular and cardiorespiratory interactions during postural stress and during mental stress [17,22].

The present work aims at combining the paradigm of Network Physiology with the algorithmic flexibility of Information Dynamics to assess the dynamical properties of the network of brain, cardiovascular and respiratory interactions during different levels of mental stress. This combined approach may help to shed light on the physiological mechanisms underlying the dynamical regulation of different organ systems during altered mental states, and may help to provide indexes with physiological meaning that can find interesting applications in contexts such as ambient assisted living scenarios [23]. Indeed, stress is a physiological condition that correlates directly to quality of life of individuals [24], and an accurate measurement of stress levels would permit implementing solutions for the prevention and treatment of such conditions. Previous attempts of studying stress mechanisms and their influence on the regulation of multiple physiological systems include the analysis of the complexity of short-term cardiovascular and respiratory signals during orthostatic and mental stress [22], the analysis of cardiorespiratory dynamics during mental arithmetic and sustained attention through bivariate entropy measures [25], and the analysis of short-term multivariate complexity of cardiovascular and respiratory dynamics under physiological stress [26]. Here, we extend the experimental protocol considering different levels of mental stress, increase the number of physiological systems analyzed simultaneously considering the cardiac, cardiovascular and respiratory systems as well as different brain subsystems, and extend the methodological approach considering univariate measures together with measures of directed coupling obtained following a fully multivariate perspective. In particular, we want to investigate how the information is stored in each node and exchanged between different nodes of the network, analyzing also the topology of such network. Moreover, the use of wearable sensors connected wirelessly favors the application of the developed framework also in a practical areas of use where stress needs to be assessed in real-life scenarios.

## 2. Information Decomposition

The theory described in this section is partially taken from [17,18]. In this study, the various physiological systems are considered as dynamic systems whose dynamical activity is described using stochastic processes. Specifically, we consider a network $X=\{X_{i}\}$ comprising the dynamic systems $X_{1},\ldots ,X_{M}$, where *M* is the number of nodes of the network (M=7 in our study). The activity of such network is described by the vector stochastic process $X=\{X_{n}\}$, where $X_{n}\;={\left[X_{1,n}\ldots X_{M,n}\right]}^{T}$ quantifies the state of the overall system at the *n*-th time step in probabilistic terms.

A realization of the stochastic process X is the multivariate time series x=[x1,x2,…,xN], containing the values measured over *N* consecutive samples that take the form of an M×N data matrix. Each row of this data matrix is a scalar time series xi=[xi,1,xi,2,…,xi,N], representing a realization of the process Xi which describes the activity of the individual dynamic system Xi. Moreover, let us consider, for example, Xj as the “target” process in our network and the remaining processes Xs, with s={1,…,M}\j, as “source” processes. Assuming that X is a Markov process of order *m*, its whole past history Xn− can be truncated using *m* lags, i.e., Xn−≅Xnm=[Xn−1T…Xn−mT]T; in a similar way, the past of the i−th scalar process can be written Xi,nm=[Xi,n−1,…,Xi,n−m]T.

A straightforward measure of the amount of information contained in the target process $X_{j}$ at time *n* is given by the entropy of the variable $X_{j,n}$:(1)$$H_{j}=H\left(X_{j,n}\right),$$
where dependence on the time index *n* can be omitted assuming stationarity of the process $X_{j}$. Then, considering the source processes, the target information can be decomposed as:(2)$$H_{j}=S_{j}+T_{j}+N_{j},$$
where $S_{j}$, $T_{j}$ and $N_{j}$ represent, respectively, the information stored in the *j*-th process, the information transferred to it from all source processes, and the new information generated about its present state when the past states of the whole network are known. Specifically, the information storage is defined in terms of the so-called self entropy as:(3)$$S_{j}=I\left(X_{j,n};X_{j,n}^{m}\right)=H\left(X_{j,n}\right)-H\left(X_{j,n}|X_{j,n}^{m}\right),$$
where I(·;·) denotes mutual information and H(·|·) denotes conditional entropy. The self entropy is a measure of the amount of information contained in the present state of the target process that can be predicted by its own past states. The total information transfer is defined in terms of the joint transfer entropy as:(4)$$T_{j}=I\left(X_{j,n};X_{s,n}^{m}|X_{j,n}^{m}\right)=H\left(X_{j,n}|X_{j,n}^{m}\right)-H\left(X_{j,n}|X_{n}^{m}\right),$$
where I(·;·|·) denotes conditional mutual information. The joint transfer entropy is a measure of the amount of information contained in the present state of the target process that can be predicted by the past states of all source processes, above and beyond the information that is predicted already by the past states of the target itself. Note that the sum of the self entropy and the joint transfer entropy of a target process quantifies the so-called prediction entropy, which is a measure of the total amount of predictive information contained in the process interpreted as a node of the considered network. The new information is defined in terms of the new generated entropy as:(5)$$N_{j}=H\left(X_{j,n}|X_{n}^{m}\right),$$
measuring the residual amount of information contained in the present state of the target process when the past states of the whole network are known. As such, the new entropy of $X_{j}$ reflects the information newly produced by the network that appears in the target process after the transition from the past states to the present state.

The measures defined above quantify the processing of information for an assigned target of the observed network of multiple interacting processes. In addition to the information received by the target from all source processes defined in Equation (Equation 4), it is possible to consider the information that the target receives referring exclusively to an individual source process. Specifically, the information transfer from the source process Xi to the target process Xj when the remaining source processes Xk, k=s\i={1,…,M}\{i,j}, are assigned, is defined in terms of the conditional transfer entropy as:(6)Ti→j|k=I(Xj,n;Xi,nm|Xk,nm,Xj,nm)=H(Xj,n|Xk,nm,Xj,nm)−H(Xj,n|Xnm).

The conditional transfer entropy is a measure of the amount of information contained in the present state of the target process that can be predicted by the past states of a specific source process, above and beyond the information that is predicted already by the past states of the target and of the other source processes. This measure reflects the information transferred between two specific processes in the network and, as such, is useful to identify the topology of the network itself.

In this study, the measures defined above were computed exploiting the linear method described in [18,27]. Under the assumption that the observed dynamical network X produces a jointly Gaussian vector stochastic process X, exact formulations can be provided as follows for the information measures defined above. The entropy of the target process $X_{j}$ is computed as [28]:(7)$$H_{j}=\ln\left(\sigma_{j}\sqrt{2\pi e}\right),$$
where $\sigma_{j}^{2}=E\left[X_{j,n}^{2}\right]$ is the variance of $X_{j}$. The new information $N_{j}$ can be instead computed as [29]:(8)$$N_{j}=\ln\left(\sigma_{j|j,s}\sqrt{2\pi e}\right),$$
where $\sigma_{j|j,s}^{2}$ is the partial variance of the target process given the past of all processes in the network, quantified as the variance of the prediction error of a linear regression of $X_{j,n}$ on $X_{n}^{m}$. In a similar way, the conditional entropy of the present of the target given its own past can be derived using linear regression as $H\left(X_{j,n}|X_{j,n}^{m}\right)=\ln\left(\sigma_{j|j}\sqrt{2\pi e}\right)$, where $\sigma_{j|j}^{2}$ is the variance of the prediction error of a linear regression of $X_{j,n}$ on $X_{j,n}^{m}$; this term, together with the information computed as in Equation (Equation 7), can be plugged in Equation (Equation 3) to compute the information storage of $X_{j}$ as:(9)$$S_{j}=\ln\frac{\sigma_{j}}{\sigma_{j|j}}.$$

Following the same reasoning, the total information transferred to the process $X_{j}$ from all sources can be computed relating the partial variance of the linear regression of $X_{j,n}$ on $X_{j,n}^{m}$ with the partial variance of the linear regression of $X_{j,n}$ on $X_{n}^{m}$:(10)$$T_{j}=\ln\frac{\sigma_{j|j}}{\sigma_{j|j,s}},$$
and the conditional information transferred from the process $X_{i}$ to the process $X_{j}$ given $X_{k}$ can be computed relating the partial variance of the linear regression of $X_{j,n}$ on $\left(X_{k,n}^{m},X_{j,n}^{m}\right)$ with the partial variance of the linear regression of $X_{j,n}$ on $X_{n}^{m}$:(11)$$T_{i\to j|k}=\ln\frac{\sigma_{j|j,k}}{\sigma_{j|j,s}}.$$

In the present study, the variance of the target process and all the partial variances that are needed for the computation of the information measures according to Equations (Equation 7)–(Equation 11) were derived using the theory of state space models as described in [30,31]. The approach is based on describing the observed network process as a vector autoregressive process, which is in turn represented as a state space model. Then, submodels are obtained from the state space model for which the state equation is obtained and the observation equation is reduced removing one or more processes. Finally, the partial variance is obtained as the variance of the prediction errors obtained regressing the present of the target process on the past of the processes which have not been removed (e.g., when the assigned source process $X_{i}$ is removed from X in the observation equation, the estimated partial variance is $\sigma_{j|j,k}$). More details about state space modeling and computation of the partial variances are given in [30,31].

Table 1 summarizes all variables and indices that will be analyzed.

## 3. Materials and Methods

In this section, the hardware configuration and the protocol for the acquisition of the physiological signals are described. The data processing of such signals for the time series extraction and the statistical analysis are also explained.

### 3.1. Hardware Configuration

The various physiological signals were acquired using low-invasive wearable devices. A t-shirt by Smartex [32] (Prato, Italy) provided the electrocardiogram (ECG) signal (lead II) and the respiratory signal, obtained with a sampling frequency of 250 Hz and 25 Hz, respectively. The respiratory signal was recorded by a piezo-resistive sensor situated at the level of the ribcage. The output is a voltage signal proportional to the sensor stretch, which was not calibrated to yield measurements of the respiratory volume in l/min. However, since our analysis was focused on the dynamic variations of the signal rather than on its absolute value, this is not an issue for the present study. The blood volume pulse (BVP) signal was recorded from photoplethysmographic sensor provided by Empatica (Milano, Italy) E4 wristband; BVP data were acquired at a sampling frequency of 64 Hz. The EEG was obtained through the Emotiv (San Francisco, CA, USA) EPOC PLUS wireless headset, which collects 14 EEG signals with a sampling frequency of 256 Hz for each channel. The various devices were connected to the same PC through a Bluetooth connection.

### 3.2. Data Acquisition

Eighteen young healthy volunteers (age: 20–30 years; gender: five females, 13 males) participated in this study. The recording sessions were performed between 10:30 a.m. and 12.00 a.m. in order to avoid differences due to day time. No caffeine had to be assumed by the participants at least three hours before a recording session. The participants were seated in front of a PC in a comfortable room at constant illumination. They were instructed to not speak, limit their movements during the recording sessions, sit comfortably without changing posture, and to try to relax before the experiment. Three different mental stress levels were induced to the participants collecting signals during specific experimental conditions. The first was a resting condition, induced by watching a relaxing video. The second was a stressful task obtained through mental arithmetic: the participants had to perform the maximum number of 3-digit addictions and subtractions in a fixed amount of time and write the solution in a text-box using the keyboard; no pen and paper were allowed as well as finger counting. The third condition was a sustained attention task induced by playing a serious game, which consisted of following a point moving on the screen using the mouse and trying to avoid some obstacles. This experimental design was devised following previous works in which varying levels of mental engagement and stress were evoked by means of nonstressful attention tasks or by stressful mental load tasks [25,33,34]. Here, we assume that playing serious games elicits a condition of sustained attention characterized by mental involvement and low stress, while performing a mental calculus is a more stressful task characterized by higher workload.

For every participant, two different recording sessions were performed (one for the mental arithmetic and one for the serious game), using the same schema (Figure 1):rest (12 min);mental arithmetic/serious game (7 min);recovery (12 min).

The two recording sessions were performed with a pause of at least 15 min between the two. All subjects were submitted all tests.

All participants provided written informed consent. The experiment was approved by the Ethics Committees of the University of Trento. The study was in accordance with the Declaration of Helsinki.

In this study, three phases of the whole protocol were considered for the analysis for each participant: the first rest phase before mental arithmetic (REST), the mental arithmetic phase (MA), and the serious game phase (SG). Here, we decided to use only the first rest phase, leaving for further studies the analysis of the recovery rest phases, to make sure that the epoch analyzed in the resting relaxed condition is free from possible habituation phenomena that could still be present after completion of the mental arithmetic stressful task.

### 3.3. Data Pre-Processing and Analysis

After simultaneous collection of the physiological signals, data were analyzed offline using MATLAB R2016b (MathWorks, Natick, MA, USA). Baseline wander of the ECG was removed using a high-pass filter with a half power frequency of 1Hz. High frequency noise was removed using a low-pass filter with a half-power frequency of 20Hz. Then, the tachogram (i.e., the sequence of the consecutive durations of the cardiac period, RR interval) was obtained detecting the R peaks in the ECG using a template matching algorithm [35,36,37], and then taking the difference between the occurrence times of consecutively detected R peaks. R peak detection is based on finding the local maxima of the cross-correlation between a template of the QRS complex and the ECG, applying a threshold on the cross-correlation, and finally locating the time of the R peak at the time of the maximum value of the aligned template [36,37]. The detected R peaks were visually inspected for the insertion of missing beats and correction of ectopic beats. From the respiration signal, a respiratory time series synchronous with the tachogram was obtained sampling the signal in correspondence of the detected R peaks [27]. The time series representative of the cardiovascular dynamics was obtained from the ECG and BVP signals calculating the sequence of the consecutive pulse arrival times (PATs); each PAT was computed as the time elapsed from the occurrence of the R peak in the ECG to the corresponding point of maximum derivative in the BVP signal, which denotes the arrival of the blood pulse at the level of the wrist [38,39,40]. Figure 2 shows schematically the detection of the points of interest for the reconstruction of the time series of RR, respiratory signal, and PAT.

The three physiological time series of RR intervals, values of the respiratory signal, and PATs were then synchronously resampled at 1Hz using spline interpolation. An example of the three resulting time series is reported in Figure 3.

For what concerns the EEG, we decided to process the signals provided from the F3 electrode. This choice was due to the fact that, in the literature, prefrontal electrodes are typically considered for the analysis of brain signals concerned with the detection of stress conditions [41,42,43]. In order to obtain time series representing the variations in time of the amplitude of the various EEG rhythms, the power spectral density (PSD) was computed on the recorded EEG signals using the periodogram method and quantifying the total power content of the δ (0.5–3 Hz), θ (3–8 Hz), α (8–12 Hz) and β (12–25 Hz) frequency bands. An example of the four time series obtained, for a single subject in the three analyzed conditions, computing the spectral power content of the EEG inside a specific band (δ, θ, α or β) is reported in Figure 4. The power spectral density was computed for EEG epochs lasting two seconds, with 50% overlap, in order to obtain one value for each bandpower at each second. The four brain time series obtained in this way, which resulted in being sampled at 1 Hz, were synchronous with those obtained resampling the three cardiovascular time series at 1 Hz. This uniformity of the final sampling rate, together with the synchronization of the signals acquired from the different devices, allowed us to obtain synchronous time series for the different body districts. The described procedure, adopted to obtain synchronous and meaningful information about the dynamics of different physiological systems, adheres to those proposed in previous studies in the field of network physiology [15,16].

With the pre-processing described above seven synchronous time series, representative of the cardiac, cardiovascular, respiratory and brain wave amplitude dynamics, were obtained with a sampling period of one second for each subject and experimental condition. Then, windows corresponding to a duration of five minutes (300 samples) were selected in each experimental condition. To reduce transient behaviors, window selection was performed after at least three minutes from the beginning of data collection in the rest periods. As regards the mental arithmetic and serious game tasks, extraction of the windows to be analyzed was performed starting from one up to two minutes after the transition from the rest phase. This choice was due to the fact that, in this case, the phenomenon of habituation can occur and therefore the earlier during the task (except the first one minute of the state change) this happens, the higher (usually) the response will be. After extracting time series of 300 points for each condition (Figure 3 and Figure 4), a restricted form of weak sense stationarity was tested using the method proposed in [44], which checks the steadiness of mean and variance across randomly selected subwindows.

The seven time series obtained from each subject and time window were interpreted as realizations of a vector autoregressive process, and the parameters of such process were estimated using the standard least squares method. The order of the underlying Markov model was identified using the Akaike Information Criterion (AIC) [45]. All the series were reduced to zero mean and unit variance. Then, considering each time series as the target $x_{j}$ (i=1,…,7), the information storage $S_{j}$, the new information $N_{j}$ and the total transfer $T_{j}$ were computed according to the methodology described in Section 2. Moreover, the conditional information transfer from the source $x_{i}$ to the target $x_{j}$ given the remaining sources $x_{k}$, $T_{i\to j|k}$ was computed for each source $x_{i}$ (i=1,…,7,i≠j).

### 3.4. Statistical Analysis

Statistical analysis was performed through a two-way analysis of variance assuming the mental state (REST, SG, MA) and the network node process (RR, RESP, PAT, δ, θ, α, β, corresponding to j=1,…,7) as categorical independent variables, and one of the information measures ($S_{j}$, $T_{j}$, $N_{j}$) as continuous dependent variable. The test was aimed at detecting statistically significant differences among the different physiological systems (network nodes) in an assigned mental state, and among the different mental states with respect to an assigned physiological system. The aim of this statistical analysis, performed at group level, is to investigate the significance of the mean differences across conditions and network nodes of each measure of information dynamics, i.e., $S_{j}$, $T_{j}$, $N_{j}$, computed for all subjects. A significance level of p=0.05 was used.

Moreover, the statistical significance of the measure of total information transfer $T_{j}$, and of the measure of conditional information transfer $T_{i\to j|k}$ computed for each source system, was assessed using a Fisher *F*-test that compares the prediction error variances of two nested linear regression models. This analysis was performed individually for each subject, for each experimental condition and target system. The conditional information transfer was considered as statistically significant, and a directed link was detected from the node *i* to the node *j* of the network, when the *F*-test returned a p<0.05.

## 4. Results

Figure 5 displays the information storage of the considered time series, reported as the distribution across subjects of the index $S_{j}$ computed in the three considered mental states, i.e., rest (REST), mental arithmetic (MA), and serious game (SG). The information stored in the cardiovascular and respiratory systems is significantly higher with respect to that stored in the four brain subsystems. Moreover, the information storage is higher for RR than for RESP, and for RESP than for PAT, the differences being statistically significant during REST and MA, and less evident during SG. Considering the transitions across the different mental states, $S_{RR}$, $S_{RESP}$ and $S_{PAT}$ show a similar trend, with a lower value of information storage in MA and in SG with respect to REST. The decrease of the information storage moving from REST to MA is documented by the statistical significance of the *F*-test for $S_{RESP}$, and by the fact that the index becomes comparable with the information storage of the EEG time series for $S_{PAT}$.

For what concerns the information storage in the EEG power time series, values of the information storage are rather low in all the three mental states, with no statistically significant differences detected among REST, MA and SG. Moreover, no statistically significant differences are present among the different EEG power series when considering the information storage evaluated in the same mental state.

Table 2 lists the median values of $S_{j}$ for every node in every state condition.

Figure 6 shows the distributions across subjects of the new information $N_{j}$ computed for each time series in the three mental states and Table 3 shows the corresponding median values. In general, the values of this quantity are complementary to those of the information storage $S_{j}$: the same average values and trends noticed in Figure 5 are here present in the opposite way. Indeed, the EEG power time series show a high amount of content of new information, while the PAT, RESP and RR generate progressively lower amounts of new information. These trends are associated with a higher complexity (lower predictability) of the brain time series, and a progressively lower complexity of cardiovascular, respiratory and cardiac time series. Also in this case, no differences can be noticed among the different mental states for what concerns the EEG time series.

Figure 7 displays the distributions of the values of total information transferred to each node of the network from all other nodes, while Table 4 lists the median values of $T_{j}$ for every node in every state condition. All nodes of the network receive a significant amount of information, as documented by the *F*-test conducted on the values of $T_{j}$ for every physiological signal and state. Indeed, the test returned a *p*-value < 0.05 in at least 15 out of 18 subjects for all target time series and in each experimental condition, documenting the existence of predictable dynamics also for the time course of the amplitude of the different EEG waves, which present a lower regularity. Figure 7 shows that the information transfer $T_{j}$ is higher for the cardiovascular and respiratory districts than for the subsystems of the brain district, with a difference that is statistically significant for the RR series in all states, for the RESP series during SG, and for the PAT series during REST and SG. For what concerns the peripheral districts (RR, RESP, PAT), the values of $T_{RR}$, $T_{RESP}$, and $T_{PAT}$ are comparable among each other during REST and MA, while, during SG, higher values of $T_{RR}$ are observed, such that the network transfers significantly higher amounts of information to RR than to RESP and PAT. The information transfer shows a statistically significant decrease moving from REST to MA for the RESP time series. As regards the brain time series, the total information transfer does not differ significantly across subsystems.

The total transfer $T_{j}$ provides an incomplete information on how the different systems are connected because it refers, for every system under consideration, only to the information that the system receives from all other systems when they are considered together. Hence, to investigate the topology of the network regarding pairwise directed interactions, an analysis of the conditional information transfer $T_{i\to j|k}$ was performed, computing the statistical significance of the estimated values of $T_{i\to j|k}$ along all possible directions, according to the analysis presented in Section 3.4. A causal link exists from one time series to another if the past of the first series helps in predicting the present of the second above and beyond the past of any other time series forming the observed network. This concept is formalized by the measure of conditional information transfer defined in Equation (Equation 6) and computed as in Equation (Equation 11). The results of such analysis are displayed in Figure 8, which shows, in each of the three mental states, the total information transferred to each node (color-coded values of the mean $T_{j}$ across subjects) and the most active connections among systems (arrows present when at least 7 subjects show significant $T_{i\to j|k}$, and thicker arrows when more than 12 subjects show significant $T_{i\to j|k}$). The absence of an arrow linking two nodes in Figure 8 suggests that the causal connection is weak or not consistent, rather than absent.

Analyzing the topology of the resulting network of brain and peripheral interactions, we can notice an evident variation in the number and location of the active connections during the different mental states. The subnetwork containing the cardiovascular and respiratory time series receives the highest amounts of information, and the active network links show that such information is transferred mostly within the nodes of this subnetwork according to a consistent topology. In particular, there is a strong bidirectional coupling between RR and RESP, a unidirectional influence from RR to PAT, and a connection between PAT and RESP. The topology of the peripheral subnetwork is the same during REST and MA, while a dominant link from RESP to PAT emerges during SG.

On the other hand, the topology of the brain subnetwork is less stable, showing bidirectional interactions between the θ and β brain wave amplitudes at REST, the emergence of α−β interactions and of multiple links directed towards the θ subsystem during MA, and a residual connectivity involving the α waves during SG. Overall, the number of connections is higher during MA than during rest, and lower during SG than during MA.

The analysis of the links connecting the nodes of the brain and peripheral subnetworks reveals important properties featuring the topology of brain–cardiovascular interactions. At REST, the high information transfer towards RR can be partly explained by the δ and β brain activities that, also interacting with each other, send information to the heart both directly acting toward RR and indirectly acting toward RESP. During MA, the connection between the cardiorespiratory network and the δ−β brain subnetwork is clearly emphasized through the presence of bidirectional interactions between δ and both RR and RESP, and the emergence of links directed from RR and from RESP towards the β node. Finally, only a residual activity of brain–heart interactions is present during SG, characterized by the connections of the pathway α→RR→θ.

## 5. Discussion

### 5.1. Information Produced and Stored in the Nodes of the Human Physiological Network

The framework of information dynamics allows for quantifying, from the analysis of the multivariate time series describing the activity of an observed physiological network, the amounts of information produced at each moment in time at the j-th node of the network, and the amount of information dynamically stored in the node. These two amounts, which are quantified by the measures of new information $N_{j}$ and of information storage $S_{j}$, reflect respectively the complexity of the node dynamics (related to the unpredictability of the present state of the target when the past network states are known) and the regularity of these dynamics (related to the predictability of the present state of the target when its own past states are known) [17]. Our results show that, in the analyzed network of brain and peripheral interactions during different stress levels, the amounts of new information and information storage are always complementary to each other (when one is higher, the other is lower), both in their variations across conditions and in their variations across network nodes. Methodologically, this behavior indicates that variations of the information content of a node (measured by the entropy $H_{j}$) and of the information transferred to it (measured by the total transfer $T_{j}$) contribute essentially to the same extent to the dynamics (see Equation (Equation 2): $N_{j}+S_{j}=H_{j}-T_{j}$). This behavior leads us to observe an invariance across physiological systems and physiological states of the capability of the network to produce and store information: if a node generates more new information, either between conditions or in comparison to another node, the information stored in it roughly decreases the same amount.

The analysis of the cardiovascular and respiratory time series reveals a decrease in regularity during the stressful cognitive tasks, reflected by the lower values of $S_{j}$ and the higher values of $N_{j}$ measured moving from REST to MA. The increased respiratory and cardiovascular complexity induced by mental stress, documented by the lower $S_{j}$ and higher $N_{j}$ during MA, is in agreement with the increase of the vascular and respiratory complexity previously found as a result of mental stress [22]. On the other hand, if considered for the cardiac system, this result is in disagreement with previous studies, which have shown that the regularity of the RR series increases significantly as a consequence of an activation of the sympathetic nervous system induced by postural stress [17,46,47]. However, the type of sympathetic activation which takes place in the case of mental stress can be related to different mechanisms than that induced by postural stress. In particular, it is likely that sympathetic activation is driven mainly by the cardiac baroreflex during postural stress, and by central commands originating in the central autonomic network during mental stress [48].

The behavior of the brain subnetwork, studied in terms of new information and information storage of the EEG power time series, is less informative in terms of the involved mechanisms. Indeed, we find, regardless of the analyzed brain subsystem and mental state, high values of $N_{j}$ and low values of $S_{j}$ which are indicative of a low predictability of the time course of the various brain wave amplitudes with respect to the cardiovascular and respiratory time series. On the other hand, the information transferred to the brain subnetwork, though remaining lower than that transferred to the peripheral subnetwork, was statistically significant. This suggests that the dynamical properties of the brain wave activity can be better understood in terms of connectivity than single node activity.

### 5.2. Information Transfer across the Nodes of the Human Physiological Network

Using an approach that is fully multivariate, we have the possibility to study how the information is transferred among the systems constituting the physiological network under consideration. In particular, the analysis of the total information transfer $T_{j}$ allows us to establish the overall amount of information received by each network node (Figure 8), while the analysis of the conditional information transfer leads to infer the topological structure through which information flows across nodes (Figure 7). In the following, we discuss our findings relevant to the analysis of information transferred within the network, distinguishing the description of the peripheral subnetwork, the brain subnetwork, and the subnetwork of brain–peripheral interactions.

The topology of the cardiovascular and respiratory subnetwork is quite consistent across conditions, showing significant cardiorespiratory and cardiovascular interactions in the three analyzed mental states. In particular, the high amounts of information transferred to the RR time series reflect a strong coupling between the heart rate variability (HRV) and respiration, which is very likely due to well established mechanisms such as the respiratory sinus arrhythmia [49] and cardiorespiratory synchronization [50]. The influences observed from RR to PAT reflect the well known effect of the heart rate on stroke volume and arterial pressure, which in turn determine variations in the PAT [51,52]. In addition, the effects of respiration on the PAT variability are expected, since they should reflect breathing influences on the intra-thoracic pressure, cardiac output, blood pressure, and ultimately blood flow velocity [52]. As a new finding, we observe that, while almost all mechanisms are stable across conditions, respiratory effects on PAT seem more elicited during the mild stress condition (SG).

Compared to the peripheral subnetwork, the brain subnetwork shows links which are less consistent (found in 7–12 subjects out of the 18) and more variable in topology across the different mental states. An apparent result is that the brain subnetwork is more connected during the stressful condition elicited by MA as compared to REST and SG. This result, which can be useful to distinguish stress states using EEG power dynamics, is consistent with studies regarding the role played by the different brain waves during rest and stressful tasks [53], and is particularly in agreement with previous reports suggesting that increased brain connectivity is expected during stressful tasks [54]. The correlation between α and β found in MA and SG is also somewhat expected, as studies in the literature report a decrease of the α power band and an increase in the θ power band during stressful and sustained attention tasks [55,56]. Moreover, the link found during MA between α and δ, which can be explained with a correlation between the two, is supported by previous findings where during mental tasks, an increase in δ and a decrease in α power was concomitantly observed [57]. Thus, our results support the hypothesis of a higher degree of connectivity within the brain network during high levels of mental stress evoked by MA. It should also be noted that participants could be more accustomed in playing video games due to their young age (from 20 to 30 years), and this may limit the capability of the designed protocol to elicit sustained attention during the SG condition. This observation could partly explain the presence of limited connectivity within the brain subnetwork during SG, and in general the higher difficulty in differentiating REST from SG observed for some of the measures considered. In this respect, the design of new experimental protocols able to elicit higher mental workloads than the SG task would be appropriate to test the hypothesis that increasing mental levels relate to higher degrees of connectivity within brain and physiological networks.

Finally, the possibility to explore brain–peripheral interactions allows us to make insightful inferences about how brain dynamics driven by central nervous system are related with autonomic effects manifested in the cardiovascular and respiratory dynamics. In this perspective, a main result is the emergence during mental stress of a higher number of network connections linking the brain dynamics (in particular, δ and β dynamics) with the cardiorespiratory subnetwork (RR and RESP nodes). This fact strengthens the hypothesis described above of a central role played by the central autonomic network during mental stress in the regulatory mechanisms. The correlation of δ and β waves with HRV was studied in several previous works, mainly related to sleep analysis. In [58,59], a correlation was found between δ EEG power and HRV, supposing an influence on the sympathetic nervous activities, while, in [60], a relationship was found between β power and the autonomic activation. In [19], the β node was the main one for the exchange of information between the cardio and brain subnetworks during the different sleep phases. Our results suggest that similar mechanisms could play a role in the increased brain–heart connectivity observed in the transition from rest to stress, possibly in analogy with the modifications observed in the transition across different sleep states. In addition, we note that the network changes substantially moving from MA to SG, with the emergence of brain–heart interaction underlined by links from α to RR and from RR to θ. The role of α and θ dynamics during sustained attention was highlighted in [61], who hypothesized that the cerebral neuronal systems producing α and θ oscillations are crucial for this task. Moreover, a correlation between cardiac autonomic activities and the θ rhythm was also found in [62] during an attention demanding meditation procedure. These results are consistent with the remodulation of the network of brain–heart interactions that we observe moving from MA to SG, possibly suggesting that different brain rhythms contribute in a different way to the link between central and autonomic activities depending on the mental state.

The presence of isolated nodes in the network represents a weak or inconsistent causal relations involving these specific network nodes. In such a case, we may hypothesize that the dynamics of the corresponding time series, like θ and α power during REST and δ power during SG, are independent of the dynamics of the other time series. Some of these behaviors are observed likely for the first time, given that the topic of our work is little explored in the literature. Taken together, the different behaviors of connectivity across brain rhythms observed in the different states contribute to characterizing the topology of the brain subnetwork in these specific states.

## 6. Conclusions

The aim of this paper was to conduct an analysis of the information dynamics of physiological networks during different levels of mental stress, in order to investigate the mechanisms underlying the processing of information that leads to complex multivariate dynamics such as those displayed by the different brain wave amplitudes on the one side, and the cardiovascular and respiratory variability on the other side. Interesting applications can also emerge in real-life scenarios thanks to the fact that wearable devices were used for the acquisition of the physiological signals. Our results indicate that a characterization of these networks is possible, both in terms of the levels of information produced and stored at each network node, and in terms of the amounts of information transferred within and between the brain and peripheral subnetworks. Such characterization was informative regarding the description of the mechanisms leading to the complexity of physiological dynamics and to the interaction between different physiological systems in a resting condition, and regarding the alteration of these mechanisms during a cognitive task evoking mental stress. The distinction between the cognitive task and the sustained attention task resulted in being more difficult, likely as a consequence of the fact that such states provoke similar autonomic changes. However, the consideration of central and autonomic effects as they are reflected in the topology of brain–heart interactions provided additional characterization among the analyzed states. Our findings can be helpful for a better comprehension of physiological mechanisms behind stress conditions.

Future studies will extend the analysis to the other electrode signals for a deeper investigation of the dynamics of the entire brain area during different levels of mental stress. Indeed, the use of the other electrodes could reveal other coupling dynamics depending on the specific electrode location, increasing the potential of the presented approach. Moreover, further analysis should focus on the comparison between the two rest phases before and after mental arithmetic or serious games to investigate possible effects remaining after the stressful tasks. Finally, it would be also important to advance the methodology adopted here investigating the presence of instantaneous (zero-lag) interactions, also in consideration of the fact that zero-lag correlations cannot be excluded among the observed physiological time series [63]. This aspect was not addressed in this study because a nontrivial modification of the estimators adopted would have been needed. For example, the direction of zero-lag interactions cannot be inferred from physiological considerations in the data analyzed in this study, and thus methods exploiting non-Gaussianity should be adopted [64,65].

## Figures and Tables

**Figure 1 entropy-21-00275-f001:**
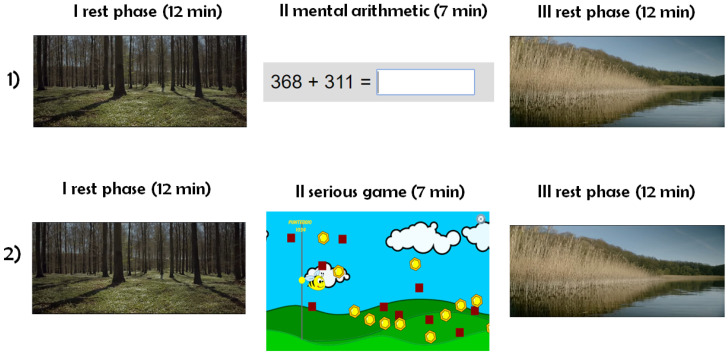
Schematic representation of the experimental protocol adopted in this study.

**Figure 2 entropy-21-00275-f002:**
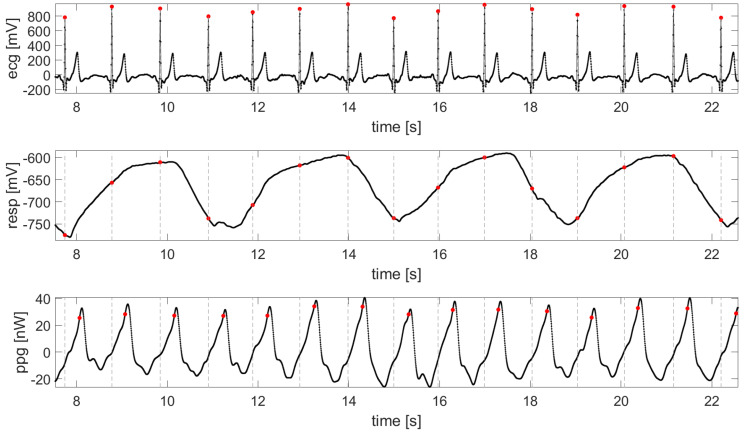
ECG, respiratory signal and BVP acquired from the wearable sensors. The red dots indicates what concerns the ECG, the detection of the R picks; for the respiratory signal, the corresponding value; and for the BVP the point of maximum derivative.

**Figure 3 entropy-21-00275-f003:**
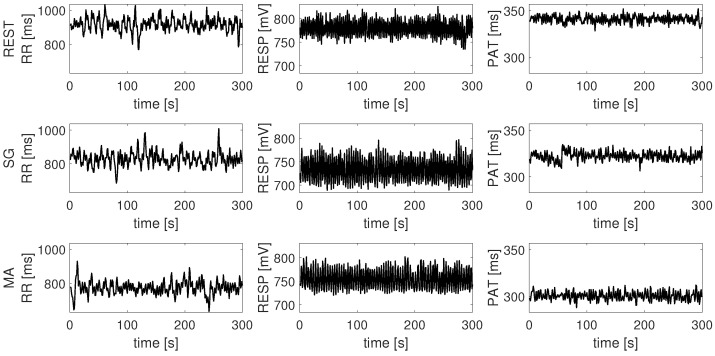
RR interval, respiratory and PAT time series measured for a representative subject during the resting phase (REST), the serious game test (SG) and the mental arithmetic test (MA).

**Figure 4 entropy-21-00275-f004:**
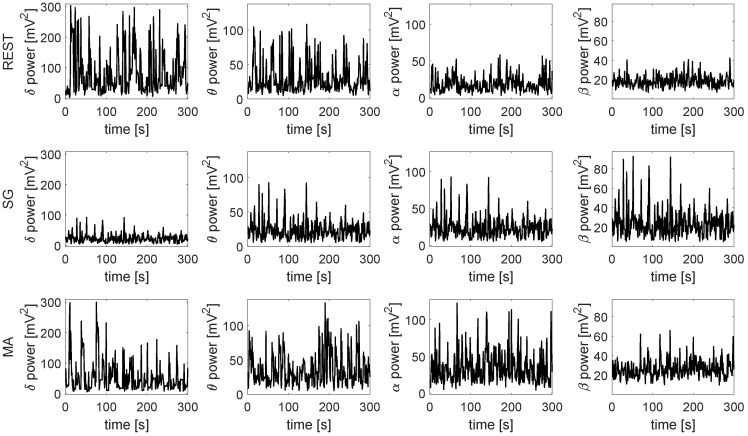
Brain wave amplitude (PSD of a sliding window of 2 s of duration with 50% overlap) time series measured for a representative subject as the time course of the δ, θ, α, β EEG power during the resting phase (REST), the serious game test (SG) and the mental arithmetic test (MA).

**Figure 5 entropy-21-00275-f005:**
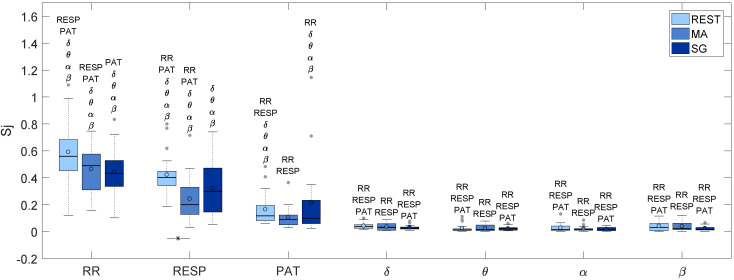
Boxplots of the information storage $S_{j}$ (*p* < 0.05) for the seven time series under consideration during rest (REST), mental arithmetic (MA), and serious game (SG). The lines under the boxplots indicate significant differences between the linked mental states as determined by the ANOVA test; moreover, the names of the time series that are significantly different from the one under consideration for any assigned mental state, listed above each boxplot.

**Figure 6 entropy-21-00275-f006:**
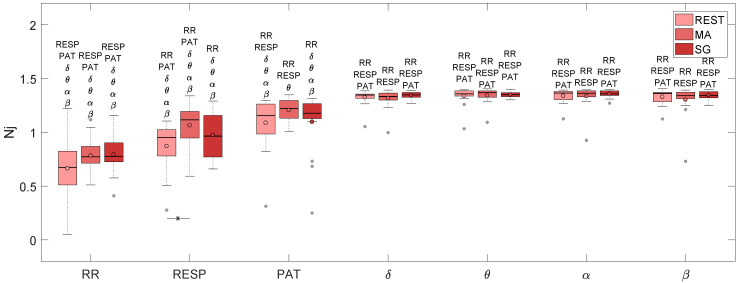
Boxplots of the new information $N_{j}$ (*p* < 0.05) for the seven time series under consideration during rest (REST), mental arithmetic (MA), and serious game (SG). The lines under the boxplots indicate significant differences between the linked mental states as determined by the ANOVA test; moreover, the names of the time series that are significantly different from the one under consideration for any assigned mental state, listed above each boxplot.

**Figure 7 entropy-21-00275-f007:**
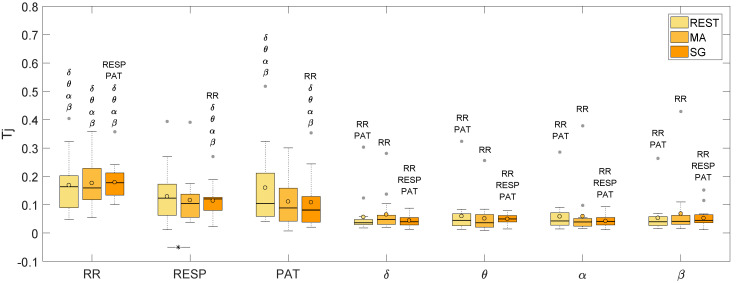
Boxplots of the total information transfer $T_{j}$ (*p* < 0.05) for the seven time series under consideration during rest (REST), mental arithmetic (MA), and serious game (SG). The lines under the boxplots indicate significant differences between the linked mental states as determined by the ANOVA test; moreover, the names of the time series that are significantly different from the one under consideration for any assigned mental state, listed above each boxplot.

**Figure 8 entropy-21-00275-f008:**
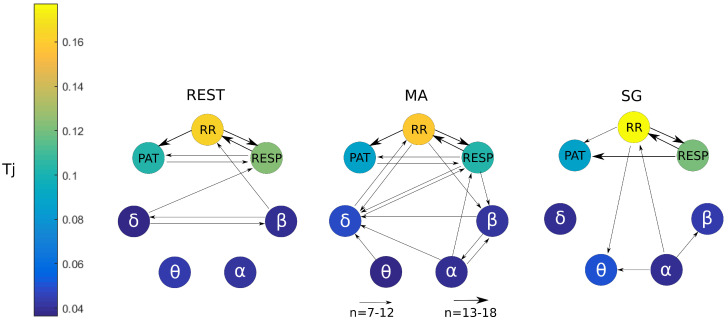
Information transfer for the cardiorespiratory-brain network using the conditional information transfer $T_{i\to j|k}$. The arrows thickness is proportional to the number of subjects for which that link is statistically significant (*p* < 0.05) using an *F*-test. The magnitude of $T_{j}$ for each node is coded accordingly to the colorbar on the left.

**Table 1 entropy-21-00275-t001:** Time series and information dynamic indices analyzed.

Time Series	Information Dynamic Indices
cardiac period (RR interval)	Information Storage ($S_{j}$)
respiration (RESP)	New Information ($N_{j}$)
pulse arrival time (PAT)	Information Transfer ($T_{j}$)
EEG $\delta_{F3}$ power	Conditional Information Transfer ($T_{i\to j|k}$)
EEG $\theta_{F3}$ power	
EEG $\alpha_{F3}$ power	
EEG $\beta_{F3}$ power	

**Table 2 entropy-21-00275-t002:** Median values of $S_{j}$ for the seven time series under consideration during rest (REST), mental arithmetic (MA), and serious game (SG).

	RR	RESP	PAT	*δ*	*θ*	*α*	*β*
**REST**	0.560	0.401	0.117	0.039	0.013	0.015	0.031
**MA**	0.490	0.200	0.088	0.032	0.014	0.016	0.022
**SG**	0.434	0.300	0.099	0.024	0.022	0.013	0.017

**Table 3 entropy-21-00275-t003:** Median values of $N_{j}$ for the seven time series under consideration during rest (REST), mental arithmetic (MA), and serious game (SG).

	RR	RESP	PAT	*δ*	*θ*	*α*	*β*
**REST**	0.674	0.951	1.155	1.347	1.357	1.365	1.362
**MA**	0.772	1.117	1.222	1.333	1.369	1.364	1.342
**SG**	0.777	0.965	1.178	1.349	1.351	1.357	1.343

**Table 4 entropy-21-00275-t004:** Median values of $T_{j}$ for the seven time series under consideration during rest (REST), mental arithmetic (MA), and serious game (SG).

	RR	RESP	PAT	*δ*	*θ*	*α*	*β*
**REST**	0.163	0.123	0.104	0.036	0.045	0.043	0.040
**MA**	0.159	0.104	0.088	0.047	0.037	0.039	0.041
**SG**	0.177	0.120	0.081	0.040	0.050	0.040	0.045

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
