# Peer review of "Information Dynamics of the Brain, Cardiovascular and Respiratory Network during Different Levels of Mental Stress"

_entropy, 2019, doi:10.3390/e21030275_

Round 1

Reviewer 1 Report

The paper investigates the potential of combining information-theoretic measures with the network physiology paradigm for the analysis of brain, cardiovascular and respiratory dynamics. This approach includes seven physiological time series (low invasive recordings) and accordingly, the associated physiological systems during different levels of mental stress. Information dynamics are studied in terms of the storage as well as new and transferred information between systems. I particularly appreciate the general concept of this follow-up paper to look at all these combined information-theoretic measures from the practical point of application to physiological systems.  

The paper is (up to minor revisions) very well structured and written (in terms of giving the theoretical background as well as methodological considerations including the coverage of related literature, technical/mathematical details of recordings, proposed methods and statistical evaluations), results are well described, discussed and summarized and sound encouraging in the field.  

Minor comments:

1) Just for completeness, it would be nice to have not only the age but also the gender distribution of the volunteers (page 5, line 144).  

2) The authors stated (and justified) to have used only signals provided from the F3 electrode. Including results of other electrodes (e.g. lateral electrode F4) would even increase the potential of future studies. The authors should add some comments about that to the discussion/conclusion section.

3) Figure 2 and 3 show sample recordings during REST, SG and MA – figure legends stated another order: REST, MA and SG – possibly to point to the assumed increasing stress level? Then it would be better to chance the order in the figures.

4) Concerning statistical analysis, it sounds like performing statistical evaluation and getting significance per single subject (page 8, line 212-217: “… for each subject…statistical significance … was assessed using…”; page 11, figure legends of figure 7: “…proportional to the number of subjects for which that link is statistically significant …” )? My understanding is that single subject would imply dependent realizations and thus a sample size of N=1?  

Author Response

The paper investigates the potential of combining information-theoretic measures with the network physiology paradigm for the analysis of brain, cardiovascular and respiratory dynamics. This approach includes seven physiological time series (low invasive recordings) and accordingly, the associated physiological systems during different levels of mental stress. Information dynamics are studied in terms of the storage as well as new and transferred information between systemsI particularly appreciate the general concept of this follow-up paper to look at all these combined information-theoretic measures from the practical point of application to physiological systems.  

The paper is (up to minor revisions) very well structured and written (in terms of giving the theoretical background as well as methodological considerations including the coverage of related literature, technical/mathematical details of recordings, proposed methods and statistical evaluations), results are well described, discussed and summarized and sound encouraging in the field.

We thank the reviewer for the careful revision of our paper and for the positive evaluation. Below we provide a point-by-point reply to the comments raised, indicating changes that were made in the text of the article.

Minor comments:

1) Just for completeness, it would be nice to have not only the age but also the gender distribution of the volunteers (page 5, line 144).

The gender distribution of the participants was included in the revised manuscript (page 6 line 149);

2) The authors stated (and justified) to have used only signals provided from the F3 electrode. Including results of other electrodes (e.g. lateral electrode F4) would even increase the potential of future studies. The authors should add some comments about that to the discussion/conclusion section.

We fully agree with this comment. Indeed, while in this paper reference to only the F3 EEG electrode was made on the basis of previous results in the literature, studying the spatial distribution of brain-brain and brain-body interactions would provide additional information and increase the potential of our approach. In this sense, we expanded the conclusion section mentioning the perspective of involving in future studies the brain time series provided by other EEG electrodes(page 16 line 466-469).

3) Figure 2 and 3 show sample recordings during REST, SG and MA – figure legends stated another order: REST, MA and SG – possibly to point to the assumed increasing stress level? Then it would be better to chance the order in the figures.

The figures were arranged ordering REST, SG and MA as these are supposed to evoke increasing levels of stress. Thanks for noting the inconsistency with the ordering of presentation in the legend, which was changed to be the same as in the figures.

4) Concerning statistical analysis, it sounds like performing statistical evaluation and getting significance per single subject (page 8, line 212-217: “… for each subject…statistical significance … was assessed using…”; page 11, figure legends of figure 7: “…proportional to the number of subjects for which that link is statistically significant …” )? My understanding is that single subject would imply dependent realizations and thus a sample size of N=1?

Most of the statistical analyses were performed at a group level investigating the significance of the mean differences across conditions and network nodes of each measure of information dynamics computed for all subjects (page 9  lines 246-249). In addition, we performed also an analysis specifically devised to assess the significance of the causal connection between two given network nodes in a given condition. This second analysis, that was based on the Fisher F-test which assesses the statistical significance of the measure of conditional information transfer, was performed individually for each subject to exploit the nature of the F-test itself which compares two linear regression model estimated from the same time series. To avoid confusion, this point was clarified in the revised paper (page 9 lines 253-254).

Reviewer 2 Report

In this work, the authors combine information theoretic measures under a network physiology and multivariate approach, applied to resting wakefulness and 2 different levels of mental stress characterization.

Network activity and connectivity are assessed in a framework of information dynamics and interactions between the different network nodes are evaluated from signals obtained using wearable sensors.

The results are promising with physiological and practical expected relevance for brain heart interactions characterization, as  claimed by the authors.

The work is very well organized and correctly written, with a solid basis stemming from previous fully reported work.

The sample seems relatively homogeneous in terms of age of the participants, which are young. This is surely important regarding the  mental stress tests that were used in this study in terms of the individual responses, In particular, the participants are maybe not very accustomed to mental arithmetic tasks and much more familiar with video games.

As the number of participants is low 18, namely when compared to the number of variables in the various analyses performed, and the expected dependence condition of the measures from each of the individuals, there can be some concerns about the statistical tests formal applicability in the analysis performed.

Anyway, in particular  the distributions reported in figures 4-6, clear indicate the expected differences in terms of the linked mental states. Maybe such insight could also be given, or at least further commented, in terms of the information  transfer, in particular for some illustrative aspect of the differences between MA and SG situations.

Author Response

In this work, the authors combine information theoretic measures under a network physiology and multivariate approach, applied to resting wakefulness and 2 different levels of mental stress characterization.

Network activity and connectivity are assessed in a framework of information dynamics and interactions between the different network nodes are evaluated from signals obtained using wearable sensors.

The results are promising with physiological and practical expected relevance for brain heart interactions characterization, as  claimed by the authors.

 The work is very well organized and correctly written, with a solid basis stemming from previous fully reported work.

We thank the reviewer for the careful revision of our paper and for the positive evaluation.

The sample seems relatively homogeneous in terms of age of the participants, which are young. This is surely important regarding the  mental stress tests that were used in this study in terms of the individual responses, In particular, the participants are maybe not very accustomed to mental arithmetic tasks and much more familiar with video games.

As the number of participants is low 18, namely when compared to the number of variables in the various analyses performed, and the expected dependence condition of the measures from each of the individuals, there can be some concerns about the statistical tests formal applicability in the analysis performed.

Anyway, in particular  the distributions reported in figures 4-6, clear indicate the expected differences in terms of the linked mental states. Maybe such insight could also be given, or at least further commented, in terms of the information  transfer, in particular for some illustrative aspect of the differences between MA and SG situations.

We agree with the reviewer in noting that the sample size can be low in an experiment like ours where many variables (network nodes) are investigated across three conditions. We also thank the reviewer for pointing out the fact that young participants can be familiarized with the serious game condition, and we agree that this may have an effect on the elicitation of the corresponding state of sustained attention. These aspects were acknowledged as possible limitations in the study design (page 6 lines 162-166) and considered in some of the illustrative descriptions of the differences between SG and MA (page 15 lines 408-417).

Reviewer 3 Report

In this work, the authors propose methods to combinate between the paradigm of Network Physiology with the information dynamics of cardiovascular, respiratory and brain systems. This reviewer considers an interesting paper to look at the response of the brain, cardiovascular and respiratory systems in function of the stress state. The authors approach a subject that is not only complex but also little studied yet, and the great unknown still, the stress. However, there are several questions and concerns that need to be addressed.

* After the mathematical description of the methods, teh authors could  introduce a table or similar, to describe all indices/parameters/variables that will be analysed.

 In Material and Methods section

*In hardware configuration, could they clarify what respiratory signal is recorded and analysed? There isn’t any reference about the acquisition system, and is unclear the type of ECG (what lead) and respiratory signal is obtained. Or if they are modified signals?

* In data acquisition – What is the basal condition of the volunteers? It is difficult to know the mental state of the participants for this study. It is unclear the conditions of the sample studied. The information most accurate of them is the age range and the hour of the record (of course relevant), but there are other relevant conditions too, that is necessary to know to evaluate the homogeneity of the sample used for the study …

* All subjects were submitted all tests?

* In data pre-processing and analysis – It is unclear the algorithms and process developed to obtain different types of time series. The algorithm presented in reference 32 (1993) is the better, the most appropriated? It would be good to introduce some comment about the validation and correction of the badly detected peaks …

* Terms such as respiration samples are imprecise in this type of work

* In Fig. 3, what is exactly that we can see? Is it the result of the sliding window of 2 s of duration with 50% overlap? Or it is a representation of the signal obtained? And what is the relation exactly,  to obtain the synchronization with cardiovascular data?

* Could they improve the description of the seven time series obtained in this process?

* In Statistical analysis – how are defined the different mental state? Especially when they differentiate between STRESS and SG (serious game?) – or is a mistake?

* It is un clear what statistical tests were used to obtain the variables with statistically significant differences

In Results section

* It is difficult to understand the process followed up for obtain these results. In the protocol there were defined two tests, differentiates by the activity developed: mental arithmetic or serious game. All two tests presented two REST phases. Now, in the results there aren’t difference between these two REST states? Are the same, probably not – because with this protocol, should interesting to understand the mental response before and after exercises.

* In all case, the magnitudes of time series are higher than magnitudes of brain parameters. Are these magnitudes comparable? It is difficult to interpreter these results. On the other hand, if additionally to the graphic representation, the authors presented numerical results these would be more understandable and interpretable.

* In the analysis of the topology of the network, which are the criteria for consider conditional information transfer?

* According to Figure 7, what is the interpretation when, in REST state there are NO two mental connection? And in SG state there aren’t one connection with delta signal?

In Discussion and Conclusion sections

* In general, the discussion presented by the authors reflect the explanation and relation between cardiovascular, cardiorespiratory system with the central nervous system – and hopefully that increase and/or decrease activity are associated.  The question is, as we can quantify this activity? Is more relevant the mental activity developed with MA or SG? Are they obtained the same results if first is MA and after SG or vice versa?

* Which are the different levels of mental stress? were there quantified?

* Sure, their findings can helpful to better understand the physiological mechanism in stress condition, but this reviewer consider that some parts should be clarified and need more formal description, before to conclude more than there are in the literature, at the moment.

Minor comments:

* line 240 – time series is repeated

* The authors could improve Figure 7 – it is impossible to read the name of parameters into dark blue connections

Author Response

In this work, the authors propose methods to combinate between the paradigm of Network Physiology with the information dynamics of cardiovascular, respiratory and brain systems. This reviewer considers an interesting paper to look at the response of the brain, cardiovascular and respiratory systems in function of the stress state. The authors approach a subject that is not only complex but also little studied yet, and the great unknown still, the stress. However, there are several questions and concerns that need to be addressed.

We thank the reviewer for the careful revision of our paper, for appreciating the novel aspects of our work, and for the questions raised, which have helped us to clarify several aspects of our work. Below we provide a point-by-point reply to the comments raised, indicating changes that were made in the text of the article.

* After the mathematical description of the methods, teh authors could  introduce a table or similar, to describe all indices/parameters/variables that will be analysed.

We inserted the table as suggested (page 5 line 130) to ease the reader in following the results.

 In Material and Methods section

*In hardware configuration, could they clarify what respiratory signal is recorded and analysed? There isn’t any reference about the acquisition system, and is unclear the type of ECG (what lead) and respiratory signal is obtained. Or if they are modified signals?

In the revised paper we clarified better that the respiratory signal was measured by a piezo-resistive sensor transducing the thoracic movements which are related to the respiratory activity (page 5 lines 136-142). The respiration signal was not calibrated to yield measurements of the respiratory volume in l/min, but this was not an issue for this study because our analysis is focused on the dynamic variations of the signal rather than on its absolute value. Moreover we clarified that the ECG was measured along the lead II. A reference was provided about the acquisition system (Smartex sensors) (page 5 line 137).

* In data acquisition – What is the basal condition of the volunteers? It is difficult to know the mental state of the participants for this study. It is unclear the conditions of the sample studied. The information most accurate of them is the age range and the hour of the record (of course relevant), but there are other relevant conditions too, that is necessary to know to evaluate the homogeneity of the sample used for the study …

In the revised paper we added details about the analyzed states, better explaining the rationale behind  the choice of the experimental conditions (also providing references), clarifying further aspects of the groups (e.g. gender) and experimental protocol (procedures followed to ensure homogeneous conditions as much as possible), and adding extra information about the basal condition of the volunteers (page 6 lines 149-154).

* All subjects were submitted all tests?

Yes, all subjects underwent the same protocol which included the execution of mental arithmetics and of serious game with resting phases before and after (see figure 1). This aspect was clarified in the revised paper (page 6 line 173).

* In data pre-processing and analysis – It is unclear the algorithms and process developed to obtain different types of time series. The algorithm presented in reference 32 (1993) is the better, the most appropriated? It would be good to introduce some comment about the validation and correction of the badly detected peaks…

In the section 3.3 we provided more details about the procedure followed to extract, from each acquired physiological signal, the seven time series used for the analysis of information dynamics. Specifically, we clarified with more detail:

(i) the template matching algorithm used to locate the QRS intervals of the ECG and extract from them the R-R intervals, also providing validation of the measured R-R interval through visual inspection with insertion of missing beats and correction of ectopic beats (page 7 lines 187-193); the algorithm presented in Ref. 32, now Ref. 35, employs standard template matching principles recalled citing other references (see new Ref. 36,37), which work fairly well in standard conditions of ECG measurement; please also note that QRS detection was not an issue in our database, thanks to the good quality of the ECG and the use of a high-pass filter removing baseline wandering (see e.g. Fig. 1);

(ii) the procedure used to extract synchronous respiratory time series by sampling the respiratory signal at the occurrence of each detected R peak in the ECG, as well as the series of the pulse arrival time by extracting fiducial points from the BVP signal related to the deflection (maximum derivative) that denotes the arrival of the blood pulse at the level of the wrist (page 7 lines 198);

(iii) the spectral analysis of the EEG signal that allowed us to derive time series representative of the variations over time of the spectral power of the EEG inside the frequency bands typically considered to analyze EEG rhythms (page 8 lines 206-213).

We inserted also Fig. 2 for a clearer explanation about how the cardio and respiratory time series were obtained.

* Terms such as respiration samples are imprecise in this type of work

We substituted the term “respiration samples” with “values of the respiratory signal” (page 8 lines 201)

* In Fig. 3, what is exactly that we can see? Is it the result of the sliding window of 2 s of duration with 50% overlap? Or it is a representation of the signal obtained? And what is the relation exactly,  to obtain the synchronization with cardiovascular data?

Fig. 3 reports the four time series obtained, for a single subject in the three analyzed conditions, computing the spectral power content of the EEG inside a specific band (delta, theta, alpha or beta). The power spectral density was computed for EEG epochs lasting 2 seconds, with 50% overlap, in order to obtain one value for each bandpower at each second. The four brain time series obtained in this way, which resulted sampled at 1 Hz, were synchronous with those obtained resampling the three cardiovascular time series at 1 Hz. This uniformity of the final sampling rate, together with the synchronization of the signals acquired from the different devices, allowed us to obtain synchronous time series for the different body districts. These aspects were expanded and better clarified in the revised manuscript (page 8 lines 212-217).

* Could they improve the description of the seven time series obtained in this process?

The improved description of the derivation of the seven time series is reported in the replies to the previous comments, and the text of the revised paper was modified accordingly. We also note that this procedure, adopted to obtain synchronous and meaningful information about the dynamics of different physiological systems, adheres to those proposed in previous studies in the field of network physiology [15,16].

* In Statistical analysis – how are defined the different mental state? Especially when they differentiate between STRESS and SG (serious game?) – or is a mistake?

We thank the reviewer for noting our mistake. In the revised paper we substituted the word “STRESS” with “MA” (page 9 lines 242) and we paid attention to uniformly mention throughout the paper that the three states considered in this study are REST, SG and MA.

* It is un clear what statistical tests were used to obtain the variables with statistically significant differences

Most of the statistical analyses were performed at a group level investigating the significance of the mean differences across conditions and network nodes of each measure of information dynamics computed for all subjects (page 9  lines 246-249). In addition, we performed also an analysis specifically devised to assess the significance of the causal connection between two given network nodes in a given condition. This second analysis, that was based on the Fisher F-test which assesses the statistical significance of the measure of conditional information transfer, was performed individually for each subject to exploit the nature of the F-test itself which compares two linear regression model estimated from the same time series. To avoid confusion, this point was clarified in the revised paper (page 9  lines 253-254).

In Results section

* It is difficult to understand the process followed up for obtain these results. In the protocol there were defined two tests, differentiates by the activity developed: mental arithmetic or serious game. All two tests presented two REST phases. Now, in the results there aren’t difference between these two REST states? Are the same, probably not – because with this protocol, should interesting to understand the mental response before and after exercises.

We thank the reviewer for this comment which led us to explain some details of the experimental protocol which were not clearly presented. The two experiments depicted in Fig. 1 were executed in two recording sessions with a pause of at least 15 min between the two. In this study, only the first rest phase for each participant was taken into consideration, leaving for further studies the analysis of the recovery rests. We decided to use only the first rest phase to make sure that the epoch analyzed in the resting relaxed condition is free from possible habituation phenomena which could still be present after completion of the mental arithmetic stressful task. These aspects were clarified in the revised paper (page 7 lines 176-181).

Nevertheless we agree with the reviewer that further studies should focus on the comparison between the two rest phases before and after mental arithmetic or serious game to investigate possible effects remaining after the stressful tasks. This interesting possibility was mentioned in the conclusions (page 16 lines 469-471)

* In all case, the magnitudes of time series are higher than magnitudes of brain parameters. Are these magnitudes comparable? It is difficult to interpreter these results. On the other hand, if additionally to the graphic representation, the authors presented numerical results these would be more understandable and interpretable.

As noted by the reviewer, the magnitude of the time series is different for the different districts. However this is not an issue for our analysis because it is focused on the dynamical predictability of the series regardless of their magnitude, and the series were reduced to zero mean and unit variance before computing the information measures (page 9 lines 235).

As regards the information measures, indeed their magnitude differs substantially between cardiovascular and brain time series. This is one of our main results, that was supported by statistical analysis as reported in the paper (page 14 lines 369-370). The interpretation of this result was reported with more detail in the revised paper, stressing that it is related to higher predictability of the cardiovascular and respiratory time series as compared to the brain time series, which reflects the presence of more regular dynamics (page 14 lines 370-373). Moreover we stressed that, although the regularity of the brain time series is lower, it was found to be statistically significant, thus documenting the existence of predictable dynamics also for the time course of the amplitude of the different EEG waves (page 11 lines 287-290). As suggested by the reviewer, tables were inserted reporting the median values of the information measures, so that to support the results of Figs. 5,6,7 (page 10-12).

* In the analysis of the topology of the network, which are the criteria for consider conditional information transfer?

The analysis of the topology of the network considers the existence of direct causal relationships between pairs of time series, which is depicted by the arrows in the graphical representation and is closely related to the concept of Granger causality between two time series. According to this concept, a causal link exists from one time series to another if the past of the first series helps in predicting the present of the second above and beyond the past of any other time series forming the observed network. In the framework of information dynamics, this concept is formalized by the measure of conditional information transfer defined in Eq. (6) and computed as in Eq. (11). This criterion used to consider the conditional information transfer was made more clear in the revised paper (page 12 lines 300-304).

* According to Figure 7, what is the interpretation when, in REST state there are NO two mental connection? And in SG state there aren’t one connection with delta signal?

Figure 7, now Figure 8, was obtained computing, for each directed link between a pair of time series, how many subjects show a statistically significant value of the conditional information transfer, and drawing an arrow only if the link was significant in more than one third of the subjects (i.e. 7 subjects). Accordingly, the absence of an arrow in the figure suggests that the causal connection is weak or not consistent, rather than absent. This aspect was better clarified in the revised paper (page 12 lines 306-310).

Nevertheless, it is also true that the presence of weak or inconsistent causal relations involving a specific network node suggests that the node is almost isolated from the others in the network. In such a case, we may hypothesize that the dynamics of the corresponding time series, like theta and alpha power during REST and delta and beta power during SG, are independent of the dynamics of the other time series. Some of these behaviors are observed likely for the first time, given that the topic of our work is little explored in the literature. Taken together, the different behaviors of connectivity across brain rhythms observed in the different states contribute to characterize the topology of the brain subnetwork in these specific states (page 16 lines 441-447).

In Discussion and Conclusion sections

* In general, the discussion presented by the authors reflect the explanation and relation between cardiovascular, cardiorespiratory system with the central nervous system – and hopefully that increase and/or decrease activity are associated.  The question is, as we can quantify this activity? Is more relevant the mental activity developed with MA or SG? Are they obtained the same results if first is MA and after SG or vice versa?

Our results indicate that some changes occur in the information processed within the brain and cardiovascular networks, as well as in the topology of such networks, moving from a resting state to a state of mental attention or mental stress. The behavior of some of the indexes (e.g., complexity of the respiration time series or number of links involving the brain subnetwork) seem to indicate that the physiological response is stronger during mental arithmetic rather than during serious game. However, we cannot provide definitive conclusions about this, also because the experiments were conducted in separate sessions and therefore any variation possibly occurring when the execution of MA and SG is switched could not be investigated. These aspects were discussed in the revised manuscript (page 6 lines 162-173).

* Which are the different levels of mental stress? were there quantified?

According to other studies in the literature (e.g. Ref. 26, 33, 34) we assume that, compared with a resting relaxed state (REST), a condition of mild mental stress can be induced by serious game playing (SG), while a task inducing more stress is the mental arithmetic (MA), in which the participants have to perform the maximum number of correct operations in a fixed amount of time. While this definition is generally accepted, we are also aware that an exact elicitation of specific mental states is difficult to achieve and may be influenced by several factors. For instance, the young participants involved in our experiment can be familiarized with the serious game condition, and this may have an effect on the elicitation of the corresponding state of sustained attention. These aspects were acknowledged as possible limitations in the study design (page 6 lines 162-166) and considered in some of the illustrative descriptions of the differences between SG and MA (page 15 lines 408-417).

* Sure, their findings can helpful to better understand the physiological mechanism in stress condition, but this reviewer consider that some parts should be clarified and need more formal description, before to conclude more than there are in the literature, at the moment.

We thank the reviewer for the positive comment, and we agree that more formal descriptions of the experimental protocol, measurement of signals and time series, as well as implementation and interpretation of the information measures were needed. We hope that the explanations provided in all the above replies and the corresponding changes in the manuscript have helped to clarify the concerns raised and to elucidate all aspects of our research.

Minor comments:

* line 240 – time series is repeated

We deleted one “time series”; thank you.

* The authors could improve Figure 7 – it is impossible to read the name of parameters into dark blue connections

We changed the text color in Fig. 7, now Fig. 8.

Reviewer 4 Report

The paper “Information Dynamics of the Brain, Cardiovascular and Respiratory Network during Different Levels of Mental Stress,” deals with the application of information-theoretic measures to seven physiological time series to test the associated physiological systems under different levels of stress. The regulation by the central and autonomic nervous systems of the cardiovascular and respiratory variability is studied in terms of the storage, new and transferred information between systems. The paper represents a follow-up of previous efforts by the authors regarding theoretical developments in information-theoretic measures; it is well-written and organized with the appropriate references. Results and interpretations are very promising. Just two questions arise as follows:

a.      Regarding the mental stress tests selected for the protocol, it seems that since the subjects are young, they are more accustomed to play video games. Consequently, the former assumption may be the reason why the network for the serious game (SG) in figure 7 is less connected than the one corresponding to MA. Then, do you think that it is possible to establish that the mental stress is less for SG than for MA for this population? If the protocol changes to something more challenge task as a chess game, then the network may be more connected.  Could the authors add comments regarding the selection of mental stress tests and the former ideas?

b.      Other efforts have pointed out the importance of instantaneous effects of other time series on the target one, and not just lagged effects. Which are your considerations about to take this issue into account for the information-theoretic approach?

Author Response

The paper “Information Dynamics of the Brain, Cardiovascular and Respiratory Network during Different Levels of Mental Stress,” deals with the application of information-theoretic measures to seven physiological time series to test the associated physiological systems under different levels of stress. The regulation by the central and autonomic nervous systems of the cardiovascular and respiratory variability is studied in terms of the storage, new and transferred information between systems. The paper represents a follow-up of previous efforts by the authors regarding theoretical developments in information-theoretic measures; it is well-written and organized with the appropriate references. Results and interpretations are very promising.

We thank the reviewer for the careful revision of our paper and for the positive evaluation. Below we provide a point-by-point reply to the comments raised, indicating changes that were made in the text of the article.

Just two questions arise as follows:

a.      Regarding the mental stress tests selected for the protocol, it seems that since the subjects are young, they are more accustomed to play video games. Consequently, the former assumption may be the reason why the network for the serious game (SG) in figure 7 is less connected than the one corresponding to MA. Then, do you think that it is possible to establish that the mental stress is less for SG than for MA for this population? If the protocol changes to something more challenge task as a chess game, then the network may be more connected.  Could the authors add comments regarding the selection of mental stress tests and the former ideas?

We thank the reviewer for pointing out the fact that young participants can be familiarized with the serious game condition, and we agree that this may have an effect on the elicitation of the corresponding state of sustained attention. These aspects were acknowledged as possible limitations in the study design (page 6 lines 162-166) and considered in some of the illustrative descriptions of the differences between SG and MA (page 15 lines 408-415). Moreover, according to the comment of the reviewer, we discussed the possibility to execute experimental protocols that elicit higher mental workloads than the SG task, and to relate these different stress levels to different degrees of connectivity within  the observed physiological network (page 15 lines 415-417).

b.      Other efforts have pointed out the importance of instantaneous effects of other time series on the target one, and not just lagged effects. Which are your considerations about to take this issue into account for the information-theoretic approach?

This is an important issue that indeed would deserve more attention in future studies. As correctly noted by the reviewer, previous research efforts have been devoted to investigate the presence of instantaneous (zero lag) interactions in networks of stochastic processes, and to quantify their impact on Granger causality measures in the time and frequency domains (e.g., see Refs. 63-65).

In fact, it would be important to incorporate instantaneous effects also in the measures of information transfer implemented in the present study, also in consideration of the fact that zero-lag correlations cannot be excluded among the observed physiological time series. While this aspect would imply a nontrivial modification of the estimators adopted in this study (also because the direction of zero-lag interactions cannot be inferred from physiological considerations in the data analyzed here, and thus methods exploiting non-gaussianity should be adopted [64,65]), the issue is surely relevant and was addressed in the revised paper (page 16 lines 471-477).